# Age-Dependent Meniscal and Chondral Damage in Eastern European Women Undergoing First-Time Knee Arthroscopy

**DOI:** 10.3390/healthcare13151822

**Published:** 2025-07-26

**Authors:** Sorin Florescu, Tudor Olariu, Daliana Ionela Minda, Diana Marian, Cosmin Grațian Damian

**Affiliations:** 1Discipline of Orthopedics-Traumatology, Department XV, “Victor Babeș” University of Medicine and Pharmacy Timișoara, Piața Eftimie Murgu 2, 300041 Timișoara, Romania; sorin.florescu@umft.ro; 2Department of Organic Chemistry, Faculty of Pharmacy, “Victor Babeș” University of Medicine and Pharmacy Timișoara, Piața Eftimie Murgu 2, 300041 Timișoara, Romania; olariu.t@umft.ro; 3Research and Processing Center for Medical and Aromatic Plants (Plant-Med), “Victor Babeş” University of Medicine and Pharmacy Timișoara, Piața Eftimie Murgu 2, 300041 Timișoara, Romania; 4Department of Pharmacognosy, “Victor Babeş” University of Medicine and Pharmacy Timișoara, Piața Eftimie Murgu 2, 300041 Timișoara, Romania; 5Department of Dentistry, Faculty of Dentistry, “Vasile Goldiş” Western University of Arad, Bulevardul Revoluției 94, 310025 Arad, Romania; marian.diana@uvvg.ro; 6Faculty of Medicine, “Vasile Goldiș” Western University of Arad, Bulevardul Revoluției 94, 310025 Arad, Romania; damian.gratian@uvvg.ro

**Keywords:** meniscal tears, chondropathy, ICRS grading, medial compartment, cartilage lesions, age-related knee injury, degenerative joint disease

## Abstract

**Background/Objectives**: This is the first study to examine age-related patterns of meniscal/chondral lesions in women undergoing first-time knee arthroscopy. **Methods**: We analyzed meniscal tear type/location and evaluated cartilage damage in femoral condyles and the tibial plateau in a medium-sized Romanian cohort (*n* = 241). **Results**: Age was associated significantly (*p* ≤ 0.004) with medial meniscal damage (O.R. = 1.04, 95% CI: 1.01–1.06), medial femoral condyle chondropathy (O.R. = 1.06, 95% CI: 1.03–1.10), and medial tibial plateau chondropathy (O.R. = 1.07, 95% CI: 1.02–1.12). Medial meniscus tear patterns differed significantly between age groups (*p* < 0.001, Cramér’s V = 0.32). Bucket-handle tears—the most common tear type—peaked in middle age (*p* < 0.001, Cramér’s V = 0.30). The two menisci showed different distributions of tear patterns in women aged ≥40 years (*p* ≤ 0.023, Cramér’s V ≤ 0.41). Meniscal tears most commonly involved the posterior third. The distribution of tear sites in menisci (medial vs. lateral) varied significantly in women aged 40–59 years (*p* = 0.020, Cramér’s V = 0.28). The medial femoral condyle and medial tibial plateau showed significant intergroup differences in ICRS scores (*p* ≤ 0.024, Cramér’s V ≤ 0.34). The frequency of ICRS grade 4 cartilage lesions increased markedly in the 40–59 age group at both sites, continuing to rise in older patients for the medial tibial plateau. **Conclusions**: Knee pathology in women worsens with age, especially in the medial compartment. Early screening (intervention) in middle-aged women may help prevent advanced joint damage.

## 1. Introduction

The knee joint is one of the most important and complex joints in the human body, with a central role in stability, movement, and weight-bearing activities [1]. The femoral condyles articulate with the tibial plateau and the patella, accommodating motion and compressive force transmission [2,3]. The tibial plateau supports the femoral condyles, distributing body weight from the femur to the tibia. Functioning as an anchorpoint for the anterior cruciate ligament, posterior cruciate ligament, medial collateral ligament, and lateral collateral ligament, the tibial plateau is also pivotal to flexion, extension, and slight rotational movements of the knee [4]. Interposed between the femoral condyles and the tibial plateau, the menisci act as shock absorbers, deepen the tibial surface, distribute loads, and prevent excessive movement [1]. These semilunar fibrocartilaginous structures are connected to the intercondylar eminence of the tibia. Moreover, the medial meniscus attaches to the medial collateral ligament and the joint capsule. This feature limits its mobility compared to the smaller lateral meniscus [3].

Sex differences in anatomy, physiology, and biomechanics of the knee joint affect joint function, injury susceptibility, and long-term musculoskeletal health. Thus, females display shorter and narrower femur and tibia, affecting the distribution of mechanical forces within the joint [4]. They possess thinner menisci, with distinct proteoglycan content and different collagen fiber organization compared to male patients [5]. In addition, women have different gait mechanics, greater knee laxity, lower cartilage volume, and weaker knee musculature [6,7,8]. There are even sex-specific metabolic differences in chondrocyte metabolism: female chondrocytes rely more on oxidative phosphorylation and male chondrocytes on glycolysis [9]. All these features render females more prone to knee injuries and osteoarthritis development and progression.

While the prevalence of knee joint injuries has been widely studied, most epidemiological research employed mixed-sex or predominantly male cohorts [10,11,12,13,14,15]. Current data derive primarily from female athletes (e.g., [16,17,18]) or specific age groups (e.g., [19,20,21]). In addition, no targeted study has comprehensively investigated the prevalence of knee joint traumas in females across the entire age spectrum. It is, however, well documented that degenerative tears are more common in females, and traumatic tears in males [22,23]. These differences stem from anatomical and hormonal differences, with feminine hormones—mainly estrogen—playing a critical role in maintaining cartilage health in women by modulating inflammation, preserving the cartilage matrix, and regulating chondrocyte activity [20,21]. There is also a substantial gap in available information on this topic in Eastern Europe and more developed countries, such as Western European countries, Japan, or the USA [24,25]. Although the mechanical and functional role of the meniscus is well established [3], the frequent co-occurrence of cartilage lesions in patients with meniscal damage is often underreported or underestimated in the literature [10]. Degeneration of the articular cartilage—particularly in the weight-bearing zones of the femoral condyle and tibial plateau—may occur concurrently or as a consequence of meniscal pathology [1,3]. This interaction between meniscal injury and chondral deterioration is especially relevant in aging female populations, where hormonal, biomechanical, and anatomical factors contribute to asymmetric patterns of joint degeneration [20,21,22]. Understanding this interplay is important since osteoarthritis is a whole-joint disease, affecting all knee structures, including meniscal degeneration and not just cartilage. Moreover, synovial inflammation correlates with joint pain and dysfunction, serving as a major risk factor for more rapid progression of structural joint deterioration in osteoarthritis [26]. Nevertheless, the age-specific patterns and anatomical distribution of such lesions—particularly in women—are yet to be fully clarified.

We aimed to analyze and characterize the prevalence and anatomical distribution of meniscal and chondral lesions across different age groups in a cohort of East European women undergoing arthroscopic intervention. Conducted at a specialized orthopedic center in the western part of Romania, the present study involved a single-site retrospective design. This setting enabled detailed intraoperative evaluation of meniscal and chondral lesions via standardized protocols and expert grading. It also captures a clinical subgroup with critical need for accurate diagnosis and intervention, namely individuals with persistent symptoms unresponsive to conservative management. We addressed meniscal lesions and concurrent chondral pathology; these cartilage-related defects are major drivers for long-term joint degeneration, including early-onset osteoarthritis. These lesions are common intra-articular knee pathologies, occurring in response to both acute trauma and chronic mechanical stress [13]. Their diagnosis and management are hence critical in preserving joint function, especially in aging populations and cohorts with limited access to specialized orthopedic care. Our focus on first-time knee arthroscopies enabled a more accurate assessment of the natural distribution and severity of meniscal and chondral lesions across age groups, ensuring that the observed patterns reflect primary pathology rather than surgical outcomes or re-injury. The present findings may serve orthopedic surgeons, sports medicine specialists, and academic researchers by improving the current understanding of knee joint pathology patterns, guiding evidence-based clinical decision-making, and supporting the development of tailored treatment strategies for East European female populations.

## 2. Materials and Methods

### 2.1. Study Population

We performed a retrospective study including female patients of all ages who underwent first-time knee arthroscopy between 1 January 2018 and 31 December 2024 at the Arad County Emergency Hospital (SCJU Arad), Clinic of Orthopaedic and Traumatology. This hospital serves a number of 150,000–200,000 patients annually. The Clinic of Orthopaedic and Traumatology—affiliated with the Faculty of Medicine from the “Vasile Goldiș” Western University of Arad (UVVG Arad)—includes over 60 beds and routinely conducts approximately 300 knee arthroscopies annually [27]. The study received approval from the Institutional Ethics Committee (IEC) of SCJU Arad (approval number 92/20.01.2025) and was run in accordance with the principles of the Declaration of Helsinki. The eligibility criteria for participant selection are summarized in Table 1, detailing the inclusion and exclusion conditions applied in the present study.

Collected variables encompassed age, type and location of meniscus tears, and type and location of chondropathy. These factors were identified as key clinical parameters due to their direct impact on injury patterns and treatment decisions/prognosis [28]. Variables (covariates) with inconsistent recording across participants were excluded, including body mass index (BMI), physical activity level, comorbidities, prior knee injuries, and occupational factors. In fact, these factors are important contributors to the development and progression of knee pathology. In particular, BMI and activity level are known to affect joint loading and cartilage wear [3,8,10,11]. However, the retrospective nature of our dataset limited the availability and completeness of these records. On the other hand, using partially missing (inconsistently) documented variables could have reduced statistical power, introduced bias, and hindered comparability with similar cohort studies. Importantly, many of these factors are highly variable across populations and difficult to standardize in surgical datasets [3,8,10]. This focus on universally recorded, anatomically grounded predictors ensured the validity, transparency, and reproducibility of our results while preventing model overfitting and maintaining analytical clarity. Due to the retrospective design, however, detailed data on symptom duration and the specific onset of meniscal lesions were not consistently available. As a result, we were unable to classify tears as acute or degenerative with certainty.

Meniscal injuries cover a diverse spectrum of pathologies, with various classification systems being used to categorize these lesions based on morphology, location, and etiology [29]. We included bucket-handle, parrot-beak, transverse (radial), and horizontal tears in this study because these lesions are the most clinically significant and diagnostically distinguishable tear types [30]. Exclusive focus on these four primary tear types maintained the statistical robustness of the study and facilitated clear comparisons between well-defined groups. The status of cartilage at femoral condyles and the tibial plateau was graded according to the International Cartilage Repair Society (ICRS) classification system: that is, grade 0: normal cartilage, no damage; grade 1: superficial lesions, issues, cracks, and indentations; grade 2: fraying, lesions extending down to <50% of cartilage depth; grade 3: partial loss of cartilage thickness, defects extending >50% of cartilage depth and possibly reaching the calcified layer; and grade 4: partial loss of cartilage thickness, defects extending >50% of cartilage depth and possibly reaching the calcified layer [31].

All patients underwent knee arthroscopy due to persistent pain, mechanical symptoms (e.g., locking or clicking), or functional limitations unresponsive to conservative treatment. Preoperative evaluation was conducted using standardized MRI protocols for all patients to identify suspected meniscal and/or chondral pathology. Intraoperative findings were thoroughly documented and used to confirm imaging-based diagnoses, ensuring consistency between preoperative imaging and surgical assessment. All procedures were performed by the same orthopedic surgical team following a standardized operative protocol, using anterolateral and anteromedial portals to enable complete visualization of the joint compartments. Interventions, including partial meniscectomy and/or chondral debridement, were performed based on lesion severity and surgeon judgment. Although formal inter-rater reliability testing was not conducted, all intraoperative lesion grading was performed by experienced orthopedic surgeons trained in the use of the ICRS classification system, thereby reducing variability in interpretation. To further minimize observer bias, lesion classification was independently reviewed by two senior orthopedic surgeons using operative notes and video documentation; any disagreements were resolved through consensus. While the retrospective nature of the study precluded blinding, the consistency of the surgical team and use of standardized imaging and classification protocols enhance the reliability and reproducibility of the findings. Informed consent was obtained from all participants.

### 2.2. Statistical Analysis

Logistic regression analysis was conducted to determine the effect of age on the prevalence of meniscus damage (any meniscus damage, medial meniscus damage, and lateral meniscus damage), associated patellar damage, femoral condyle chondropathy (medial femoral condyle chondropathy and lateral femoral condyle chondropathy), and tibial plateau chondropathy (medial tibial plateau chondropathy and lateral tibial plateau chondropathy). Any meniscus damage was operationalized as a binary variable, indicating the presence of at least one meniscal lesion identified during arthroscopy. The patients were then divided into three age groups: group 1, <40 years; group 2, 40–59 years; and group 3, ≥60 years. The rationale behind this stratification consisted of two main components. First, these age groups are clinically relevant. Thus, patients younger than 40 years often present to hospital with traumatic or sports-related meniscal tears [10,13]. In contrast, middle-aged females aged 40–59 years represent a transitional group when degenerative changes begin to occur in the context of aging and menopause [28]. Older adults aged ≥60 years, on the other hand, display predominantly degenerative meniscal tears and osteoarthritis [32]. Second, this stratification approach ensured a sufficient number of cases for robust comparisons since dividing 241 patients into more strata could reduce statistical power and increase variability [33]. This simple age-based model accounts for biomechanical/degenerative factors while keeping the analysis meaningful.

Frequency analysis was applied on all outcomes, not only on outcomes significantly associated with age in the logistic regression model. This broad approach was chosen to capture trends that logistic regression might miss, especially in cases with borderline *p*-values or complex interactions [34]. Chi-square (χ^2^) tests with 3 × 2 contingency tables served to identify differences in the proportion of females with and without meniscus/patella damage [35]. When these tests yielded significant results, we performed planned pairwise comparisons with Chi-square tests between adjacent age groups. More precisely, we compared younger patients (<40 years) with middle-aged patients (40–59 years), and the latter strata with older participants (≥60 years). This stepwise procedure allowed us to identify age-related trends in the prevalence of meniscal and chondral damage. To correct for multiple testing, a Bonferroni correction was applied for these comparisons [36]. A standard significance threshold of *p* ≤ 0.05 was used for overall comparisons, while a more stringent threshold of *p* ≤ 0.025 (0.05/2) was applied for the planned pairwise tests. This dual-threshold strategy was selected to balance sensitivity and statistical rigor.

The same algorithm was implemented for analysis of the meniscal tear location and cartilage damage on the femoral condyles and tibial plateau. In addition, Chi-square tests were utilized to assess intragroup differences, such as assessing differences between medial and lateral compartment damage within each age category. To assess effect size, Cramér’s V was calculated for all multi-category contingency tables (e.g., 3 × 4 or 3 × 5), including meniscal tear types, anatomical tear locations, and ICRS grading distributions. This parameter is a measure of effect size, determining the strength of association between categorical variables and the magnitude of observed differences. This approach was not applied on any meniscus damage since this composite variable aggregates medial and lateral lesions already analyzed separately; including it in further analysis would result in statistical redundancy. According to the Cramér’s V interpretation guidelines, values between 0.10 and 0.19 are considered weak, 0.20–0.59 moderate, and ≥0.60 strong associations [37]. Based on data from the statistical literature related to exploratory studies, we used a Cramér’s V ≥ 0.20 as a threshold to identify a meaningful size effect [38,39]. All analyses were conducted using Statistica 10 software [40].

## 3. Results

Table 2 shows the results of logistic regression analysis, with meniscal/chondral lesions as outcomes and age as the predictor. The association between age and AMD was statistically significant, with each one-year increase in age resulting in an increase of 3% in the odds of having meniscal damage. This effect was primarily attributable to the medial meniscus; each one-year increase in age corresponded to a significant 3% rise in the odds of medial meniscus damage. In contrast, no significant associations were found for lateral meniscus damage or any patellar damage.

A significant association was observed between age and medial femoral condyle chondropathy, with each additional year increasing the likelihood of this pathology by 6%. A similar trend was found for the chondropathy of the medial tibial plateau. However, no significant relationships were identified for lateral femoral condyle chondropathy or lateral tibial plateau chondropathy.

Table 3 shows the distribution of different types of meniscal tears and associated patellar lesions in the study population. We identified significant differences in the prevalence of any meniscal damage between different age categories (Chi-square test, χ^2^ = 10.51, *p* = 0.006). The prevalence of any meniscus damage increased significantly from younger to middle-aged females (Chi-square test, χ^2^ = 8.64, *p* = 0.009) and then declined in older patients, nearing statistical significance (Chi-square test, χ^2^ = 2.95, *p* = 0.086). A similar age-dependent pattern was observed for medial meniscal damage (Chi-square test, χ^2^ = 10.26, *p* = 0.006, Cramér’s V = 0.27), supporting a moderate distributional shift. Its frequency peaked in females aged 40–59 years compared to younger individuals (Chi-square test, χ^2^ = 9.01, *p* = 0.002, Cramér’s V = 0.27), and remained elevated in the older females but without reaching significance (Chi-square test, χ^2^ = 1.27, *p* = 0.260, Cramér’s V = 0.14). These results reflect a clinically relevant change in the prevalence of medial meniscal tears with age, especially during midlife. In contrast, no significant differences existed for lateral meniscal damage (Chi-square test, χ^2^ = 3.17, *p* = 0.204, Cramér’s V = 0.15) or any patella damage (Chi-square test, χ^2^ = 2.41, *p* = 0.299, Cramér’s V = 0.13).

Table 4 shows the distribution of different types of meniscal tears (and their location) in the study population. Bucket-handle tears were the most common type of meniscal tears for both menisci. We note the elevated occurrence of parrot-beak tears in the medial meniscus of women younger than 40 years of age and those aged 60 years and above. The prevalence of tear patterns affecting the medial meniscus differed significantly across age groups (Chi-square test, χ^2^ = 33.10, *p* < 0.001, Cramér’s V = 0.32), suggesting a moderate effect size and a clear age-driven divergence. The frequency of bucket-handle tears increased in middle-aged females (Chi-square test, χ^2^ = 11.28, *p* < 0.001, Cramér’s V = 0.30) before declining in older females to levels comparable to those observed in younger patients (Chi-square test, χ^2^ = 0.79, *p* = 0.375, Cramér’s V = 0.37). Although the latter comparison did not reach statistical significance, the relatively strong effect sizes from these two comparisons indicate a potentially meaningful shift in tear patterns, likely reflecting anatomical and degenerative changes associated with aging.

There were no significant differences in the distribution of lateral meniscus tear patterns across age groups (Chi-square test, χ^2^ = 3.17, *p* = 0.204, Cramér’s V = 0.19). However, we note an evident trend of age-related increase in the prevalence of bucket-handle tears. Moreover, the distribution of tear patterns in the medial meniscus and lateral meniscus differed significantly in subjects aged 40–59 years (Chi-square test, χ^2^ = 9.57, *p* = 0.023, Cramér’s V = 0.31). This indicates a moderate shift in the location and nature of damage across compartments in midlife. A similar pattern was observed in females aged 60 years or over (Chi-square test, χ^2^ = 11.69, *p* = 0.008, Cramér’s V = 0.41), where the stronger effect size provides evidence for a disproportionate impact of aging on the medial meniscus. In contrast, no significant differences were found in younger women (Chi-square test, χ^2^ = 3.46, *p* = 0.321, Cramér’s V = 0.21). Parrot-beak tears were the primary drivers of these differences (Table 3).

The posterior third was the most common location of meniscus tears (Table 4). Both menisci revealed no significant age-related variation in the location of meniscus tears (Chi-square tests, χ^2^ ≤ 5.31, *p* ≥ 0.257, Cramér’s V ≤ 0.18). When comparing the intragroup distribution of tear sites in the lateral meniscus versus medial meniscus, significant differences were found in females aged 40–59 years (Chi-square test, χ^2^ = 7.78, *p* = 0.020, Cramér’s V = 0.28), but not in those younger than 40 years of age (Chi-square test, χ^2^ = 5.14, *p* = 0.076, Cramér’s V = 0.26) or aged 60 years and older (Chi-square test, χ^2^ = 2.11, *p* = 0.347, Cramér’s V = 0.17).

Figure 1 presents the distribution of different types of knee-related chondropathies in the study population. We observed significant intergroup differences in ICRS scores for medial femoral condyle chondropathy (Chi-square test, χ^2^ = 17.63, *p* = 0.024, Cramér’s V = 0.24), but not for lateral femoral condyle chondropathy (Chi-square test, χ^2^ = 3.19, *p* = 0.992, Cramér’s V = 0.32). At the level of the medial femoral condyle, the most notable change was the sharp increase in the frequency of ICRS grade 4 cartilage lesions in the middle-aged cohort versus younger patients (Chi-square test, χ^2^ = 8.19, *p* = 0.004, Cramér’s V = 0.25), reflecting a moderate progression in cartilage damage severity. Females 60 years and older showed a similar distribution of different ICRS classes to that seen in patients aged 40–59 years (Chi-square test, χ^2^ = 8.60, *p* = 0.072, Cramér’s V = 0.10) despite showing a higher prevalence of ICRS grade 3 cartilage lesions. The incidence of advanced chondropathy was significantly elevated in the medial femoral condyle compared to the lateral condyle, irrespective of age category (Chi-square tests, χ^2^ ≤ 3.91, *p* ≤ 0.048, Cramér’s V ≤ 0.45). This observation hints at a moderate to strong association between the anatomical location and the incidence of advanced chondropathy.

At the level of the medial tibial plateau, chondropathy severity also differed significantly between age groups (Chi-square test, χ^2^ = 28.44, *p* < 0.001, Cramér’s V = 0.34), indicating a moderate effect size. No significant differences were detected in the case of the lateral tibial plateau (Chi-square test, χ^2^ = 8.01, *p* = 0.433, Cramér’s V = 0.18). Planned comparisons yielded significant results (Chi-square tests, χ^2^ ≥ 5.41, *p* ≤ 0.021, Cramér’s V ≤ 0.43), revealing an evident trend of increase in the prevalence of ICRS grade 4 cartilage lesions with age (Table 4). The largest increase occurred from the young to the middle-aged female cohort. In patients aged 60 years and above, intragroup analysis revealed a significant difference in the distribution of ICRS scores between the medial and lateral tibial plateau (χ^2^ = 8.28, *p* = 0.004), with a moderate effect size (Cramér’s V = 0.37). The observed differences were largely driven by the prevalence of advanced cartilage lesions, which occurred more than threefold more often in the medial tibial plateau among the oldest patients (Table 4). This indicates a more severe chondropathy in the medial compartment of the tibial plateau. However, no significant differences were observed for the other age strata (Chi-square test, χ^2^ ≤ 0.72, *p* ≥ 0.397, Cramér’s V ≥ 0.15).

## 4. Discussion

To the best of our knowledge, this study is the first to comprehensively examine knee joint-related injuries in a diverse female-only cohort undergoing arthroscopy in Eastern Europe. The contribution of our work to the field of orthopedics is threefold. First, it clarifies transitional patterns around menopause (40–59 years), supporting the view that hormonal and structural changes significantly alter knee biomechanics and vulnerability to injury in females. Second, this investigation provides scientists with a detailed morphological and anatomical characterization of tear types and anatomical location—uncommon in observational cohort analyses with substantial patient enrollment. Third, our results reinforce the notion that weight-bearing zones (medial femoral condyle, medial tibial plateau) are especially vulnerable in aging female knees—a point with implications for preventive strategies and surgical planning.

Age was significantly associated with injuries affecting the medial meniscus, medial femoral condyle, and medial tibial plateau, with no associations seen in the lateral compartment. One can hence expect medial-sided knee injuries to primarily account for the age-dependent increase in the likelihood of any meniscal injuries in females. Clinical data support the stronger effect of aging on the medial compartment of the knee compared to the lateral compartment [11,41,42,43]. Increased weight-bearing forces during activities such as walking, standing, stair climbing, and other routine functional movements contribute to the greater vulnerability of the medial knee compartment to osteoarthritis and other degenerative changes [41]. Medial compartment full-thickness cartilage defects also progress faster in older adults than lateral defects due to various factors: e.g., knee varus alignment, increased weight, and the presence of bipolar defects—chondral or osteochondral damage on both articulating surfaces of a joint [44]. In fact, osteoarthritis is nearly ten times more common in the medial compartment than in the lateral compartment, reflecting the greater stress and degeneration experienced by the knee’s medial side [45,46].

The prevalence of medial meniscus damage increased from younger to middle-aged females (40–59 years), followed by a decline in older patients (≥60 years). Several interacting anatomical, biomechanical, and clinical factors may account for this pattern. Females aged 40 to 59 are typically in perimenopause or early postmenopause. This period is marked by a decrease in ovarian function, triggering a progressive reduction in estrogen and progesterone output. These hormonal shifts (associated with aging) are major drivers of knee osteoarthritis in women, with multiple mechanisms modulating the interface between female sex steroid signaling and the pathophysiology of this condition. The presence of estrogen receptors (ERα and ERβ) in articular chondrocytes reveals the importance of this hormone in modulating cartilage at the molecular level. Estrogen inhibits inflammation and cellular senescence/apoptosis while protecting chondrocytes from associated degenerative processes. This endocrine factor stimulates proteoglycan and collagen synthesis by chondrocytes and enhances the expression of cartilage-specific genes [47,48]. Ovariectomized rats on estrogen therapy show attenuated cartilaginous degeneration, supporting its protective role against osteoarthritis progression [49]. Estrogen also promotes fibrocartilage chondrogenesis and inhibits degeneration in female mice via estrogen receptor alpha (ERα), partly by modulating the Wnt signaling pathway and protease activity [50]. However, this protective action of estrogen on existing cartilage could be offset by its inhibitory effect on cartilage regeneration. For example, it decreases the expression of cartilage-specific genes such as type II collagen and aggrecan in adipose-derived stem cells [51]. The role of progesterone—the other key female hormone—in the knee joint is underexplored, but evidence supports its potential protective role in maintaining knee joint cartilage volume [52]. This imbalance in key ovarian hormones is likely to weaken meniscal tissue, increase susceptibility to microtrauma, and accelerate early degenerative changes in the medial meniscus, contributing to the peak in medial meniscus pathology during this transitional age.

In parallel, aging is associated with reduced muscle strength, proprioceptive decline, and changes in joint loading patterns—all of which alter knee biomechanics [6,20,21,22]. These changes may increase medial compartment stress and shear forces on the meniscus, particularly during daily activities such as stair climbing and walking. The combination of declining hormonal protection and biomechanical overload may explain the sharp rise in medial meniscus tears and advanced chondropathy in middle-aged women. Although this age group often remains physically active (e.g., work, exercise, household tasks) [53], age-related decline in muscle strength and proprioception cannot be fully compensated for. This mismatch between physical demand and joint stability predisposes them to degenerative or mixed-type meniscus lesions. In contrast, females older than 60 years typically have lower activity levels, limiting exposure to mechanical stress and acute injury risk [54]. While hormonal decline plateaus in these women, decreased physical activity may partially reduce loading forces—potentially explaining the lower prevalence of meniscal tears but continued progression of cartilage lesions, particularly in weight-bearing zones like the medial tibial plateau. Older patients with severe degeneration are also more likely to receive non-operative management rather than arthroscopy [55]. This category may hence be underrepresented in surgical datasets despite presenting with meniscal degeneration.

Bucket-handle tears occur when the inner margin of the meniscus separates from the peripheral body, forming a displaced longitudinal tear that resembles a bucket handle [56]. This type of cartilage injury was the most prevalent in both menisci. These findings are in line with data from orthopedic practice, with reported incidence surpassing 30% [57,58]. The differing frequencies of these lesions across the menisci—with a peak in middle age for the medial meniscus and an age-dependent increase for the lateral meniscus—may be related to several anatomical, biomechanical, and age-related determinants. The medial meniscus is tightly connected to the joint capsule and tibial plateau and less mobile, and thus more prone to shear stress during rotational activities [1,3]. Meniscal integrity may be further compromised by hormonal changes in perimenopause [20,21,22,48]. On the other hand, the pattern of lateral meniscus tears likely reflects cumulative microtrauma and age-related degeneration [1,5,11,44].

Parrot-beak tears were the second most common type of meniscal lesions. These present an inwardly displaced free-edge flap forming a curved configuration reminiscent of a parrot’s beak [58]. These lesions exhibited an age-dependent, bimodal trend in the medial meniscus, peaking in the youngest and oldest strata. These findings lend support to a dual mechanism: trauma in the youngest females and degeneration in the elderly females. Given their rarity in the lateral meniscus and the lack of an evident age-related pattern, one can expect that the unique anatomical and biomechanical features of the lateral compartment provide a protective effect against these types of tears. Medical evidence supports this assumption. The lateral meniscus is more circular and covers a larger portion of the tibial plateau compared to the medial meniscus. These features allow it to move more during joint motion and distribute load more evenly. During knee flexion and rotation, the lateral meniscus can also shift posteriorly by up to 10–12 mm; the medial meniscus, by contrast, moves only 2–5 mm due to its firm capsular attachments [59]. This greater excursion further aids in dissipating shear stress, protecting the lateral meniscus from the oblique radial forces that typically produce parrot-beak morphology [2,3,11,60].

Most tears were located in the posterior third of the meniscus. This is in agreement with the findings of previous studies, with the prevalence of this location reaching up to 77% of all meniscus tears [61,62]. This vulnerability of the posterior horn to injury originates from its anatomical and functional characteristics, i.e., limited mobility, high mechanical loading during flexion, poor vascularity, and key role in weight-bearing and joint stabilization [63].

Our data reveal the complex patterns of knee chondropathy and the marked age-dependent increase in its prevalence. The steep rise in advanced cartilage lesions of the medial femoral condyle in middle-aged women shows that degenerative processes at this site accelerate during perimenopause. Consistent with the literature data [64,65,66], we detected a clinically relevant tendency toward severe cartilage degeneration with increasing age. In fact, Bikash et al. estimates that over 80% of individuals over 50 years develop features of degenerative cartilage changes in the knee [1]. In addition, Katano and colleagues report a significantly higher incidence of medial femoral cartilage defects in the 70–79 age group versus younger women [66].

The prevalence of ICRS grade 4 lesions in older women (≥60 years) versus the middle-aged cohort differed across the medial knee compartment: it continued to increase for the medial tibial plateau, but leveled off for the medial femoral condyle. The medial femoral condyle is exposed to higher load-bearing forces during daily activities (e.g., walking, stair climbing) compared to its lateral counterpart. This key component of the knee joint is therefore likely to exhibit an earlier onset of severe cartilage degeneration [67] and show more advanced lesions in older women. On the other hand, the medial tibial plateau may show a more progressive pattern of degeneration, with cumulative mechanical stress and age-related cartilage wear continuing into older age [68]. Additional factors may also contribute to these progression patterns, e.g., differences in subchondral bone remodeling and cartilage repair capacity in femoral and tibial surfaces.

Chondropathy severity increased with age, and the medial compartment—including both the medial femoral condyle and the medial tibial plateau—was more affected than the lateral compartment. It is known that the former is more prone to cartilage degeneration and osteoarthritis due to higher joint loading in normal knee alignment [62]. In fact, this knee compartment is the most common site of cartilage defects [65,68,69]. On the other hand, the medial tibiofemoral compartment often shows the earliest and most advanced wear [70]. The lateral compartment tends to exhibit less degeneration unless specific risk factors (e.g., valgus alignment, lateral meniscus injury) are present [69]. Interestingly, risk factors have a compartment-specific effect; for example, elevated BMI is associated with more extensive cartilage damage in the medial and patellofemoral compartments, but not in the lateral compartment [71]. These data align with the lack of significant differences in the distribution of ICRS scores between the lateral femoral condyle and the lateral medial plateau. This renders the articular cartilage in the lateral compartment of the knee less vulnerable to degenerative forces that come with age and weight.

Meniscal tears frequently coexist with cartilage defects, and this combination is associated with worsened clinical outcomes, including increased pain, reduced joint stability, and accelerated progression of osteoarthritis [10,12]. Menisci play a critical role in load transmission and joint preservation; hence, their damage can trigger a cascade leading to cartilage degradation and altered biomechanics [1]. Recent clinical studies have shown that patients undergoing arthroscopy for meniscal pathology often present with coexisting chondral lesions, which may contribute to poorer functional recovery and long-term joint deterioration [26,70]. These findings highlight the need to consider meniscal and chondral lesions not as isolated events, but as interconnected pathologies that influence prognosis and therapeutic planning.

Taken together, the present findings reveal a complex pattern in the development of meniscal and chondral damage among female patients. Women under 40 years showed the lowest prevalence of meniscal and chondral lesions, and parrot-beak tears were common in the medial meniscus. Cartilage damage was generally mild, with ICRS grades 0–2 predominating in femoral condyles and the tibial plateau. Medial meniscus damage peaked in the midlife women (40–59 years) together with the prevalence of bucket-handle tears. This group exhibited the most pronounced increase in ICRS grade 4 lesions at the medial femoral condyle and medial tibial plateau. Moderate effect sizes occurred most often in this age cohort. Tear patterns differed significantly between compartments, indicating a critical phase of structural degeneration. Although the prevalence of meniscal tears declined in older women (≥60 years), severe cartilage lesions remained frequent, especially in the medial tibial plateau. Significant compartmental differences (medial vs. lateral) persisted, emphasizing the progression of degenerative chondropathy with age.

The current study used a sex-specific and age-inclusive framework, with a mature female-only cohort composed of young, perimenopausal, and elderly women. The present findings demonstrate an evident age-related pattern of degeneration affecting the medial compartment of the knee—the area most exposed to load-bearing forces [68,69]. This study also provides a comprehensive mapping of meniscal tear types and chondropathy grades. These insights may have direct implications for preventive care and surgical prioritization. However, their translation into routine clinical practice poses serious challenges. Thus, clinicians cannot precisely identify at-risk individuals before advanced damage occurs due to the lack of standardized screening protocols. Symptomatology varies widely; some patients with advanced lesions remain asymptomatic, whereas others with minor findings report substantial pain [1,3]. In addition, the timing of surgical intervention (versus conservative management) in middle-aged and older women is not well defined. Moreover, the current primary care approaches do not address the midlife degenerative changes in women, thereby delaying targeted prevention or referral.

The identification of middle age (40–59 years) as a high-risk window for structural knee damage supports the development of preventive strategies such as neuromuscular training, load management, targeted screening during the perimenopausal period, early intervention in middle-aged women with medial compartment involvement, or early referral for orthopedic evaluation. In older women (≥60 years), where meniscal tears may be less frequent but cartilage degeneration persists, our data may help guide decisions toward conservative treatment (e.g., bracing, physical therapy, viscosupplementation) versus surgical intervention, depending on symptom severity, lesion location, and functional goals. We note that the multifactorial nature of osteoarthritis—especially in aging women—necessitates preventive and therapeutic strategies that address not only hormonal and biomechanical influences but also individual pain processing profiles [71,72]. To build on these results, future studies should incorporate functional outcome measures (e.g., WOMAC, KOOS, gait analysis) to correlate structural damage with clinical performance and quality of life. The potential protective role of hormone therapy on cartilage health and meniscal integrity also warrants investigation in prospective longitudinal cohorts, particularly those capturing the menopausal transition. Moreover, upcoming studies should take into account computational modeling and biomechanics to advance our understanding of joint degeneration mechanisms in women. In fact, data-driven deep learning approaches have already been used to predict ligament fatigue failure and assess risk factors underlying musculoskeletal deterioration, and hence precede (accompany) degenerative changes in meniscal and chondral structures [73].

Like any observational study, our investigation has potential drawbacks. First, this study involved a retrospective single-site design, a moderate sample size, and only patients who underwent arthroscopy. This approach limits causal inference, control over confounders, and generalizability of the results to other populations with different genetics or healthcare access. It may also introduce selection bias by excluding individuals managed conservatively, potentially diverging from patterns typically seen in broader population studies. However, retrospective single-site cohort studies are routinely used to identify clinically relevant patterns in real-world settings as they provide high internal validity and consistency in data collection [74]. In addition, homogeneous surgical cohorts allow direct assessment and classification of intra-articular pathology while limiting the impact of confounding factors. Second, we excluded potentially relevant variables, such as BMI, occupation, and physical activity levels. This deliberate methodological decision aimed to avoid overfitting, ensure consistency across cases, and improve reproducibility, enabling robust statistical modeling under resource-limited conditions [75]. Third, we did not include postoperative or long-term clinical outcomes, such as functional recovery, pain levels, or progression to osteoarthritis. Nonetheless, this study was designed to assess prevalence and anatomical patterns, not therapeutic outcomes. Fourth, our analysis was limited by the absence of reliable data on symptom duration and tear chronicity. This drawback restricts our ability to determine the potential influence of prolonged meniscal damage on the severity of associated cartilage lesions. On the other hand, the primary focus of our study was to document anatomical distribution and age-related prevalence, not to assess causality or progression. Therefore, while symptom duration is a relevant variable, its absence does not invalidate the descriptive and comparative findings derived from these data. Finally, we recruited only women who underwent arthroscopy. This could have resulted in the exclusion of patients managed conservatively (non-operatively), resulting in a disproportionate prevalence of more severe pathology. Arthroscopy provides direct visualization and confirmation of intra-articular pathology, thereby enhancing diagnostic accuracy compared to imaging alone [76]. This study hence provides high-fidelity data on true lesions requiring surgical intervention. In addition, this study also considered the magnitude of differences across age groups by reporting effect sizes using Cramér’s V, a standard measure of association strength for categorical data. Several moderate to strong effect sizes were identified—particularly for patterns of medial meniscus damage and chondropathy severity—highlighting the clinical relevance of these findings beyond *p*-values. By quantifying the strength of associations, this approach ensures a more nuanced understanding of age-related shifts in intra-articular knee pathology.

## 5. Conclusions

This retrospective study provides a comprehensive analysis of age-related patterns in meniscochondral pathology among Eastern European women undergoing first-time knee arthroscopy. Our findings reveal a significant age-associated increase in the prevalence of degenerative lesions affecting the medial knee compartment—specifically the medial meniscus, medial femoral condyle, and medial tibial plateau. Tear patterns of the medial meniscus, but not those of the lateral meniscus, differed significantly between age groups. The bucket-handle tears were the most common tear type, peaking in middle-aged women (40–59 years). The two menisci revealed significantly different distributions of tear patterns in this group and the older cohort (≥60 years), with parrot-beak tears being the primary drivers of these differences. Across all cases, tears most frequently involved the posterior third of the meniscus. Notably, the distribution of tear locations between the medial and lateral menisci differed significantly in the 40–59-year age group (*p* = 0.020). Chondropathy severity, assessed via ICRS grading, plateaued in the middle-aged cohort for the medial femoral condyle, but continued to rise with age in the medial tibial plateau, reflecting divergent biomechanical loading patterns and degenerative dynamics. These findings underscore the importance of sex-specific and age-targeted approaches in orthopedic care. In particular, the perimenopausal period (40–59 years) emerged as a critical window of vulnerability for structural knee degeneration. Clinicians should consider early screening and preventative strategies during this transitional phase to delay or mitigate joint deterioration. Moreover, the observed compartment-specific degeneration supports tailored surgical planning and emphasizes the need for medial compartment preservation in aging female patients.

## Figures and Tables

**Figure 1 healthcare-13-01822-f001:**
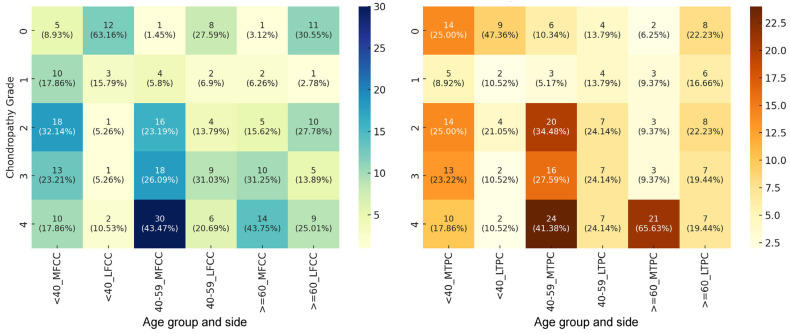
Heatmap showing the age-stratified distribution of chondral lesions associated with meniscal tears for the femoral condyle (**left**) and tibial plateau (**right**). Data are given as absolute values with the corresponding percentages (in parentheses). Age groups were defined as follows: <40—patients younger than 40 years; 40–59—patients aged between 40 and 59 years; ≥60—patients aged 60 years and older. MFCC, medial femoral condyle chondropathy; LFCC, lateral femoral condyle chondropathy; MTPC, medial tibial plateau chondropathy; LTPC, lateral tibial plateau chondropathy; 0, ICRS grade 0 (normal cartilage, no damage); 1, ICRS grade 1 (superficial lesions, issues, cracks, and indentations); 2, ICRS grade 2 (fraying, lesions extending down to <50% of cartilage depth); 3, ICRS grade 3 (partial loss of cartilage thickness, cartilage defects extending down >50% of cartilage depth as well as down to calcified layer); 4, ICRS grade 4 (complete loss of cartilage thickness).

**Table 1 healthcare-13-01822-t001:** Inclusion and exclusion criteria for participant selection.

Inclusion Criteria	Exclusion Criteria
Women aged 18 years or older	Prior knee surgery
Confirmed diagnosis of meniscal tears (with or without patellar damage) via MRI/arthroscopy	Ligament injuries or complex knee trauma
First-time arthroscopic intervention	Neuromuscular disorders affecting knee stability (e.g., multiple sclerosis, Parkinson’s disease)
Presence of persistent knee symptoms unresponsive to conservative treatment (e.g., pain, locking, catching, or functional limitation)	Uncontrolled chronic conditions affecting joint health (e.g., diabetes, cardiovascular disease with mobility limitation)
Complete medical records including age, sex, diagnosis, type/location of meniscal tear, and knee chondropathy location	Inflammatory arthritis or systemic joint diseases
	Knee malformations or malalignments
	Post-traumatic or fracture-related arthroscopy
	Concomitant ligament reconstruction procedures

**Table 2 healthcare-13-01822-t002:** Effect of age on the odds of meniscal, patellar, and chondral knee injuries in females.

Variable	β	SE	Wald z	*p*-Value	OR (95% CI)
AMD	0.03	0.01	2.26	0.024 *	1.03 (1.00; 1.05)
MeMD	0.03	0.01	2.88	0.004 **	1.04 (1.01; 1.06)
LaMD	−0.01	0.02	−0.58	0.557	0.99 (0.95; 1.02)
PaD	0.03	0.02	1.85	0.064	1.03 (0.98; 1.07)
MFCC	0.07	0.01	3.73	<0.001 ***	1.06 (1.03; 1.10)
LFCC	0.03	0.04	0.81	0.414	1.03 (0.95; 1.11))
MTPC	0.07	0.02	3.06	0.002 **	1.07 (1.02; 1.12)
LTPC	0.04	0.03	1.19	0.233	1.04 (0.97; 1.11)

β, beta coefficient; SE, standard error; Wald z, Wald z-statistic; OR (95% CI), odds ratio with 95% confidence interval; AMD, any meniscus damage; MeMD, medial meniscus damage; LaMD, lateral meniscus damage; PaD, any patellar damage; MFCC, medial femoral condyle chondropathy; LFCC, lateral femoral condyle chondropathy; MTPC, medial tibial plateau chondropathy; LTPC, lateral tibial plateau chondropathy. Odds ratios are given with lower and upper bounds of the 95% confidence interval (in parentheses). Marked values (*) denote statistically significant odds ratios (Wald test, ***—*p* < 0.001, **—*p* < 0.01, *—*p* < 0.05).

**Table 3 healthcare-13-01822-t003:** Age-stratified distribution of meniscal tear types and their location.

Age Range	n	AMD		MeMD		LaMD		PaD	
		Yes	No	Yes	No	Yes	No	Yes	No
<40 years	75	26 (34.7%)	49 (65.3%)	20 (26.7%)	55 (73.3%)	9 (12.0%)	66 (88.0%)	12 (16.0%)	63 (84.0%)
40–59 years	98	31 (63.3%)	18 (36.7%)	27 (55.1%)	22 (44.9%)	8 (16.3%)	41 (83.7%)	4 (8.2%)	45 (91.8%)
≥60 years	68	6 (35.3%)	11 (64.7%)	6 (35.3%)	11 (64.7%)	0 (0.0%)	17 (100.0%)	1 (5.9%)	16 (94.1%)

AMD, any meniscus damage; MeMD, medial meniscus damage; LaMD, medial meniscus damage; PaD, any patellar damage. Data are given as absolute values with the corresponding percentages (in parentheses).

**Table 4 healthcare-13-01822-t004:** Age-stratified distribution of meniscal tearmorphology.

	<40 Years		40–59 Years		≥60 Years	
Meniscus Tear Type						
	MeMD	LaMD	MeMD	LaMD	MeMD	LaMD
BH	23 (41.08%)	10 (52.62%)	50 (72.46%)	17 (58.62%)	10 (31.25%)	23 (63.89%)
PB	18 (32.14%)	2 (10.53%)	3 (4.35%)	2 (6.90%)	16 (50.00%)	5 (13.89%)
TT	10 (17.85%)	5 (26.34%)	13 (18.84%)	3 (10.34%)	5 (15.63%)	5 (13.89%)
HT	5 (8.92%)	2 (10.51%)	3 (4.35%)	7 (24.14%)	1 (3.12%)	3 (8.33%)
Meniscus Tear Site						
	MeMD	LaMD	MeMD	LaMD	MeMD	LaMD
1/3 A	4 (7.15%)	2 (10.52%)	5 (7.24%)	4 (13.79%)	5 (15.62%)	9 (25.00%)
1/3 M	8 (14.28%)	7 (36.85%)	11 (15.49%)	11 (37.94%)	3 (9.38%)	6 (16.16%)
1/3 P	44 (78.57%)	10 (52.63%)	53 (7.87%)	14 (48.27%)	24 (75.00%)	21 (58.33%)

BH, bucket-handle tears; PB, parrot-beak tears; TT, transverse tears; HT, horizontal tears; MeMD, medial meniscal damage; LaMD, lateral meniscal damage; 1/3 A, anterior third of meniscus; 1/3 M, middle third of meniscus; 1/3 P, posterior third of meniscus. Data are given as absolute values with the corresponding percentages (in parentheses).

## Data Availability

All the data generated or analyzed in this study are included in this published article.

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
