# Peer review of "Age-Dependent Meniscal and Chondral Damage in Eastern European Women Undergoing First-Time Knee Arthroscopy"

_healthcare, 2025, doi:10.3390/healthcare13151822_

Round 1

Reviewer 1 Report

Comments and Suggestions for Authors

Thank you for the opportunity to review your manuscript. This is a well-executed and valuable contribution to the understanding of sex- and age-specific knee pathology. The following suggestions aim to refine and enhance your work.

General Comments:

  • The article is clearly written, well-organized, and grounded in solid clinical rationale.
  • The inclusion of an all-female cohort and stratification by age is a significant strength.
  • Consider expanding briefly on the translational relevance of these findings in the Introduction or Discussion (e.g., implications for early interventions, rehabilitation, or public health).

Specific Comments:

Title:

  • Consider specifying the population in the title, e.g., “...in Eastern European Women Undergoing First-Time Arthroscopy” for clarity and indexing.

Abstract:

  • Include effect sizes or 95% confidence intervals along with p-values for main results.
  • The conclusion could mention the clinical implication of early detection or intervention in middle-aged women.

Introduction:

  • Lines 50–54: Expand on the “pediatric triad” (if relevant to previous studies) or clarify this term. It may be a mislabeling if referring to adult female pathology.
  • Strengthen the rationale for excluding BMI and other covariates by briefly acknowledging their known role but clarifying data limitations.

Methods:

  • Lines 85–89: Improve the structure of inclusion/exclusion criteria for readability. Consider using bullet points or a summary table.
  • Clarify whether the arthroscopies were performed by the same surgical team and if inter-rater reliability was assessed for lesion classification.
  • Indicate if blinding was used for outcome assessment and how potential bias was mitigated.

Results:

  • Table 1: Add confidence intervals for all odds ratios and verify that decimal notation is consistent.
  • Figures 1 and 2: Include brief legends clarifying what * and ** denote.
  • Clarify the terms “G3” and “G4” in the main text when referring to ICRS grades (first mention).
  • Include a brief summary of significant findings for each age group to improve readability.

Discussion:

  • Lines 321–325: Expand on hormonal changes and biomechanics as potential explanatory factors for age differences. You’ve mentioned them, but further interpretation could strengthen the argument.
  • Consider a short paragraph on implementation challenges for screening or prevention programs based on these findings.
  • You may also emphasize how this evidence could inform criteria for surgical vs. conservative treatment planning in older women.

Conclusion:

  • Consider highlighting how these findings could be translated into sex-specific preventive care strategies.
  • Suggest future studies that could include functional outcome measures or explore hormone therapy effects.

Tables and Figures:

  • Ensure that all statistical symbols and abbreviations are defined in the table/figure legends.
  • Standardize decimal notation (e.g., 0.05 vs .05).
  • Add legends that define significance markers and abbreviations directly in the figures.

Reference Suggestions:

  • Add the following references to contextualize the importance of individualized strategies in knee care:
    • Osteoarthritis: a call for research on central pain mechanism and personalized prevention strategies.
    • Effects of Orthopedic Manual Therapy on Pain Sensitization in Patients with Chronic Musculoskeletal Pain: An Umbrella Review
  • Please note that any papers recommended in the report are for reference purposes only and are not mandatory. You are welcome to cite and reference other relevant papers related to this topic.
Comments on the Quality of English Language
  • Overall, the manuscript is well-written. Minor edits are needed for preposition use, article use (“a” vs. “the”), and sentence structure (e.g., subject-verb agreement).
  • Carefully proofread for minor formatting inconsistencies, such as spacing between references and punctuation around citations.

Author Response

Dear Reviewer,

We would like to sincerely thank you for the time and effort spent reviewing our manuscript. We appreciate the insightful and constructive comments, which have helped us to significantly improve the quality and clarity of our work.

Below, we provide a detailed, point-by-point response to each of the reviewers' comments. All changes made in the revised manuscript have been highlighted. We hope that the revised version of our manuscript now meets your expectations. We remain at your disposal for any further clarifications.

Thank you again for your insightful feedback.

#Reviewer 1

Yes

Can be improved

Must be improved

Not applicable

Does the introduction provide sufficient background and include all relevant references?

(x)

( )

( )

( )

Is the research design appropriate?

(x)

( )

( )

( )

Are the methods adequately described?

( )

(x)

( )

( )

Are the results clearly presented?

( )

(x)

( )

( )

Are the conclusions supported by the results?

(x)

( )

( )

( )

Are all figures and tables clear and well-presented?

(x)

( )

( )

( )

General Comments:

The article is clearly written, well-organized, and grounded in solid clinical rationale. The inclusion of an all-female cohort and stratification by age is a significant strength. Consider expanding briefly on the translational relevance of these findings in the Introduction or Discussion (e.g., implications for early interventions, rehabilitation, or public health).

Q1. Consider specifying the population in the title, e.g., “...in Eastern European Women Undergoing First-Time Arthroscopy” for clarity and indexing.

R1: We have changed the article title to reflect the pertinent suggestion fo the reviewer; it reads as :” Age-Dependent Meniscal and Chondral Damage in Eastern European Women Undergoing First-Time Knee Arthroscopy”

Q2. Abstract:Include effect sizes or 95% confidence intervals along with p-values for main results.

The conclusion could mention the clinical implication of early detection or intervention in middle-aged women.

R2: We have revised the Abstract to include both effect sizes (odds ratios) and their corresponding 95% confidence intervals for the main statistically significant results derived from our logistic regression models. Specifically, the updated text in the abstract section now reads :

Age was significantly associated (p ≤ 0.004) with medial meniscal damage (O.R. = 1.04, 95% CI: 1.01–1.06), medial femoral condyle chondropathy (O.R. = 1.06, 95% CI: 1.03–1.10), and medial tibial plateau chondropathy (O.R. = 1.07, 95% CI: 1.02–1.12).

In addition, we have computed Cramér’s V as a measure of effect size for variables compared across age groups using chi-square tests. In the revised version of the manuscript we have included details related the values of chi-square and the corresponding Cramér’s V values for these tests. We have used a Cramér’s V ≥ 0.30 as a threshold to identify a meaningful size effectbased on the data from statistical literature (see newly introduced references 36, 37, and 38 for justification). This revised part in the Statistical analysis section now reads:

The same algorithm was implemented for analysis of the meniscal tear location and cartilage damage on the femoral condyles and tibial plateau. In addition, Chi-square tests were utilized to assess intragroup differences, such as assessing differences between medial and lateral compartment damage within each age category. To assess effect size, Cramér’s V was calculated for all multi-category contingency tables (e.g., 3×4 or 3×5), including meniscal tear types, anatomical tear locations, and ICRS grading distributions. To assess effect size, Cramér’s V was calculated for all multi-category contingency tables (e.g., 3×4 or 3×5). This parameter is a measure of effect size, determining the strength of association between categorical variables and the magnitude of observed differences. This approach was not applied on any meniscus damage since this composite variable aggregates medial and lateral lesions already analyzed separately; including it in further analysis would result in statistical redundancy. According to the Cramér’s V interpretation guidelines, values between 0.10–0.19 are considered weak, 0.20–0.59 moderate, and ≥ 0.60 strong associations [36]. Based on data from statistical literature related to exploratory studies, we used a Cramér’s V ≥ 0.20 as a threshold to identify a meaningful size effect [37,38]. All analyses were conducted using Statistica 10 software [39].

Besides reporting effect sizes (Cramér’s V) in the Results section, and brief interpretative notes were added where relevant to clarify the magnitude and significance of distributional differences. e.g.:

We note the elevated occurrence of parrot beak tears in the medial meniscus of women younger than 40 years of age and those aged 60 years and above. The prevalence of tear patterns affecting the medial meniscus differed significantly across age groups (Chi-square test, χ² = 33.10, p < 0.001, Cramér’s V = 0.32), suggesting a moderate effect size and a clear age-driven divergence.”

“Moreover, the distribution of tear patterns in medial meniscus and lateral meniscus differed significantly in subjects aged 40-59 years (Chi-square test, χ² = 9.57, p = 0.023, Cramér’s V = 0.31). This indicates a moderate shift in the location and nature of damage across compartments in midlife. A similar pattern was observed in females aged 60 years or over (Chi-square test, χ² = 11.69, p = 0.008, Cramér’s V = 0.41), where the stronger effect size provides evidence for a disproportionate impact of aging on the medial meniscus.”

For more details please the section Results from the revised version of our manuscript. We have also introduced the effect sizes (as the values of Cramér’s V) in the abstract for significant data—as suggested by the reviewer. Moreover we have rewritten the conclusions part of the abstract to address the reviewer’s requests; it now reads:

Conclusions: Knee pathology in women worsens with age, especially in the medial compartment. Early screening (intervention) in middle-aged women may help prevent advanced joint damage.

Q3. Introduction:

    Lines 50–54: Expand on the “pediatric triad” (if relevant to previous studies) or clarify this term. It may be a mislabeling if referring to adult female pathology.

    Strengthen the rationale for excluding BMI and other covariates by briefly acknowledging their known role but clarifying data limitations.

R3: We thank the reviewer for the observation. Upon review, we confirm that the term “pediatric triad” does not appear in the manuscript. This may have been a misattribution, and no clarification was necessary.

While variables such as BMI are known to play an important role in joint health and disease progression, we deliberately excluded them from the analysis due to inconsistent or incomplete recording across participants. Although their relevance is well established in the literature, the extent of missing data would have introduced bias and reduced the reliability of the multivariable models. Therefore, to preserve the integrity of the analysis, only covariates with consistently recorded values were retained. This rationale is detailed in the revised version of the manuscript; please see the second half of the first paragraph from subsection 2.1. Study Population. The revised text now reads :

Collected variables encompassed age, type and location meniscus tears, and type and location of chondropathy. Our study included these factors as key clinical parameters due to their direct impact on injury patterns, treatment decisions, and prognosis [27]. These variables ensured statistical robustness, clinical relevance, and comparability with existing research [1,2,27]. We deliberately excluded variables (covariates) with inconsistent recording across participants, including body mass index (BMI), physical activity level, comorbidities, prior knee injuries, and occupational factors. Indeed, these factors are important contributors to the development and progression of knee pathology. In particular, BMI and activity level are known to affect joint loading and cartilage wear [3,8,10,11]. However, the retrospective nature of our dataset limited the availability and completeness of these records. On the other hand, using partially missing (inconsistently) documented variables could have reduced statistical power, introduced bias, and hindered comparability with similar cohort studies. Importantly, many of these factors are highly variable across populations and difficult to standardize in surgical datasets [3,8,10]. This focus on universally recorded, anatomically grounded predictors ensured the validity, transparency, and reproducibility of our results while preventing model overfitting and maintaining analytical clarity.”

Q4. Lines 85–89: Improve the structure of inclusion/exclusion criteria for readability. Consider using bullet points or a summary table.

R4. The appropriate changes have been conducted in revised version of the manuscript; we used a summary table- see Table 1.

Q5. Clarify whether the arthroscopies were performed by the same surgical team and if inter-rater reliability was assessed for lesion classification.

    Indicate if blinding was used for outcome assessment and how potential bias was mitigated.

R5. Yes. We have clarified the issues in the revised version of the manuscript; please see the last and fourth paragraph from subsection 2.1. Study Population:

All patients underwent knee arthroscopy due to persistent pain, mechanical symp-toms (e.g., locking or clicking), or functional limitations unresponsive to conservative treatment. Preoperative evaluation was conducted using standardized MRI protocols for all patients to identify suspected meniscal and/or chondral pathology. Intraoperative findings were thoroughly documented and used to confirm imaging-based diagnoses, ensuring consistency between preoperative imaging and surgical assessment. All pro-cedures were performed by the same orthopedic surgical team following a standardized operative protocol, using anterolateral and anteromedial portals to enable complete vis-ualization of the joint compartments. Interventions, including partial meniscectomy and/or chondral debridement, were performed based on lesion severity and surgeon judgment. Although formal inter-rater reliability testing was not conducted, all in-traoperative lesion grading was performed by experienced orthopedic surgeons trained in the use of the ICRS classification system, thereby reducing variability in interpretation. To further minimize observer bias, lesion classification was independently reviewed by two senior orthopedic surgeons using operative notes and video documentation; any disagreements were resolved through consensus. While the retrospective nature of the study precluded blinding, the consistency of the surgical team and use of standardized imaging and classification protocols enhance the reliability and reproducibility of the findings. Informed consent was obtained from all participants.”

Q6. Results:

Table 1: Add confidence intervals for all odds ratios and verify that decimal notation is consistent.

Figures 1 and 2: Include brief legends clarifying what * and ** denote.

Clarify the terms “G3” and “G4” in the main text when referring to ICRS grades (first mention).

Include a brief summary of significant findings for each age group to improve readability.

R6. "We have checked and corrected the decimal notation to ensure consistency throughout the Table 2 (former Table 1). We have also clarified what * and ** denote.

We also appreciate the reviewer’s observation. However, the terms “G3” and “G4” were not used in the manuscript. Throughout the text, we referred to cartilage damage using the full ICRS grade descriptions (e.g., "ICRS Grade 3" or "advanced chondropathy") to ensure clarity. We have double-checked the manuscript to confirm that abbreviated forms such as “G3” or “G4” do not appear.

We have also included a brief summary of significant findings for each age group to enhance clarity and improve the readability of the results section. These summaries highlight key differences in the prevalence and patterns of meniscal and chondral lesions across the three age categories (<40 years, 40–59 years, and ≥60 years); please see the antepenultimate paragraph from the Discussion section. The updated text now reads:

“ Taken together, the present findings reveal a complex pattern in the development of meniscal and chondral damage among female patients. Women under 40 years showed the lowest prevalence of meniscal and chondral lesions and parrot-beak tears were common in the medial meniscus. Cartilage damage was generally mild, with ICRS grades 0–2 predominating in femoral condyles and tibial plateau. Medial meniscus damage peaked in the midlife women (40–59 years) together with the prevalence of bucket-handle tears. This group exhibited the most pronounced increase in ICRS grade 4 lesions at the medial femoral condyle and medial tibial plateau. Moderate effect sizes occurred most often in this age cohort. Tear patterns differed significantly between compartments, indicating a critical phase of structural degeneration. Although the prevalence of meniscal tears declined in older women, severe cartilage lesions remained frequent, especially in the medial tibial plateau. Significant compartmental differences (medial vs. lateral) persisted, emphasizing the progression of degenerative chondropathy with age.

Q7. Discussion:Lines 321–325: Expand on hormonal changes and biomechanics as potential explanatory factors for age differences. You’ve mentioned them, but further interpretation could strengthen the argument.

R7. We have expanded the Discussion section to provide a more detailed interpretation of how hormonal decline and biomechanical changes during aging contribute to the observed age-specific patterns of meniscal and chondral degeneration. These explanatory mechanisms help contextualize the peak in structural damage among women aged 40–59 years and the sustained severity of cartilage lesions in older patients. The updated text now reads; please see the third and the fourth paragraphs from the Discussion section:

The prevalence of medial meniscus damage increased from younger to middle-aged females (40–59 years) followed by a decline in older patients (≥ 60 years). Several inter-acting anatomical, biomechanical, and clinical factors may account for this pattern. Females aged 40 to 59 age are typically in perimenopause or early postmenopause. This period is marked by a decrease in ovarian function, triggering a progressive reduction in estrogen and progesterone output. These hormonal shifts (associated with aging) are major drivers of knee osteoarthritis in women, with multiple mechanisms modulating the interface between female sex steroid signaling and the pathophysiology of this condition. The presence of estrogen receptors (ERα and ERβ) in articular chondrocytes reveals the importance of this hormone in modulating cartilage at the molecular level. Estrogen inhibits inflammation and cellular senescence/apoptosis while protecting chondrocytes from associated degenerative process. This endocrine factor stimulates proteoglycan and collagen synthesis by chondrocytes and enhances the expression of cartilage-specific genes [47,48]. Ovariectomized rats on estrogen therapy show attenuated cartilaginous degeneration, supporting its protective role against osteoarthritis progression [49]. Estrogen also promotes fibrocartilage chondrogenesis and inhibit degeneration in female mice via estrogen receptor alpha (ERα), partly by modulating the Wnt signaling pathway and protease activity [50]. However, this protective action of estrogen on existing cartilage could be offset by its inhibitory effect on cartilage regeneration. For example, it decreases the expression of cartilage-specific genes such as type II collagen and aggrecan in adipose-derived stem cells [51]. The role of progesterone—the other key female hormone—in the knee joint is underexplored, but evidence supports its potential protective role in maintaining knee joint cartilage volume [52]. This imbalance in key ovarian hormones is likely to weaken meniscal tissue, increase susceptibility to micro-trauma, and accelerate early degenerative changes in the medial meniscus, contributing to the peak in medial meniscus pathology during this transitional age.

In parallel, aging is associated with reduced muscle strength, proprioceptive decline, and changes in joint loading patterns—all of which alter knee biomechanics [6,20,21,22]. These changes may increase medial compartment stress and shear forces on the meniscus, particularly during daily activities such as stair climbing and walking. The combination of declining hormonal protection and biomechanical overload may explain the sharp rise in medial meniscus tears and advanced chondropathy in middle-aged women. Although this age group often remains physically active (e.g., work, exercise, household tasks) [53], age-related decline in muscle strength and proprioception cannot be fully compensated for. This mismatch between physical demand and joint stability predisposes them to degenerative or mixed-type meniscus lesions. In contrast, females older than 60 years typically have lower activity levels, limiting exposure to mechanical stress and acute injury risk [54]. While hormonal decline plateaus in these women, de-creased physical activity may partially reduce loading forces—potentially explaining the lower prevalence of meniscal tears but continued progression of cartilage lesions, par-ticularly in weight-bearing zones like the medial tibial plateau. Older patients with severe degeneration are also more likely to receive non-operative management rather than arthroscopy [55]. This category may hence be underrepresented in surgical datasets de-spite presenting with meniscal degeneration.

Q8. Consider a short paragraph on implementation challenges for screening or prevention programs based on these findings. You may also emphasize how this evidence could inform criteria for surgical vs. conservative treatment planning in older women.

R8. We have now included two dedicated paragraphs discussing the implementation challenges for screening and prevention programs, emphasizing how our findings may inform clinical decision-making. The new paragraphs are found at the end of the Discussion section; see the the antepenultimate and penultimate paragraphs.

“The current study used a sex-specific and age-inclusive framework, with a mature female-only cohort composed of young, perimenopausal, and elderly women. The present findings demonstrate an evident age-related pattern of degeneration affecting the medial compartment of the knee—the area most exposed to load-bearing forces [68,69]. This study also provides a comprehensive mapping of meniscal tear types and chondropathy grades. These insights may have direct implications for preventive care and surgical prioritization. However, their translation into routine clinical practice poses serious challenges. Thus, clinicians cannot precisely identify at-risk individuals before advanced damage occurs due to the lack standardized screening protocols. Symptomatology varies widely; some patients with advanced lesions remain asymptomatic, whereas others with minor findings report substantial pain [1,3]. In addition, the timing of surgical intervention (versus conservative management) in middle-aged and older women is not well defined. Moreover, the current primary care approaches do not address the midlife degenerative changes in women, thereby delaying targeted prevention or referral.

The identification of middle age (40–59 years) as a high-risk window for structural knee damage supports the development of preventive strategies such as neuromuscular training, load management, targeted screening during the perimenopausal period, early intervention in middle-aged women with medial compartment involvement, or early referral for orthopedic evaluation. In older women (≥ 60 years), where meniscal tears may be less frequent but cartilage degeneration persists, our data may help guide decisions toward conservative treatment (e.g., bracing, physical therapy, viscosupplementation) versus surgical intervention, depending on symptom severity, lesion location, and func-tional goals. We note that the multifactorial nature of osteoarthritis—especially in aging women—necessitates preventive and therapeutic strategies that address not only hor-monal and biomechanical influences but also individual pain processing profiles [71,72]. To build on these results, future studies should incorporate functional outcome measures (e.g., WOMAC, KOOS, gait analysis) to correlate structural damage with clinical per-formance and quality of life. The potential protective role of hormone therapy on cartilage health and meniscal integrity also warrants investigation in prospective longitudinal cohorts, particularly those capturing the menopausal transition.

Q9. Conclusion:Consider highlighting how these findings could be translated into sex-specific preventive care strategies. Suggest future studies that could include functional outcome measures or explore hormone therapy effects.

R9. We have made the necessary improvements in the revised version of the manuscript; please see the penultimate paragraph of the Discussion section above at R9.

Q10. Tables and Figures:

Ensure that all statistical symbols and abbreviations are defined in the table/figure legends.

Standardize decimal notation (e.g., 0.05 vs .05).

Add legends that define significance markers and abbreviations directly in the figures.

R10. Decimal notation has been standardized throughout the manuscript and figures (e.g., using “0.05” instead of “.05”). In addition, legends for all figures/tables now include clear explanations of significance markers (e.g., p < 0.05, p < 0.01, p < 0.001) and define all abbreviations used.

Q11. Reference Suggestions:

Add the following references to contextualize the importance of individualized strategies in knee care:

Osteoarthritis: a call for research on central pain mechanism and personalized prevention strategies.

Effects of Orthopedic Manual Therapy on Pain Sensitization in Patients with Chronic Musculoskeletal Pain: An Umbrella Review.

R11. These references have been added in the revised version of our manuscript; see R9 and references 70,71.

Q12. Comments on the Quality of English Language

Overall, the manuscript is well-written. Minor edits are needed for preposition use, article use (“a” vs. “the”), and sentence structure (e.g., subject-verb agreement).

Carefully proofread for minor formatting inconsistencies, such as spacing between references and punctuation around citations.

R12. We have carefully reviewed and edited the manuscript to address minor issues related to article and preposition usage, sentence structure (including subject–verb agreement), and overall grammar. We have also corrected formatting inconsistencies—such as spacing and punctuation around references and in-text citations—throughout the manuscript to ensure consistency and clarity.

Reviewer 2 Report

Comments and Suggestions for Authors

My comments are as follows: 

Introduction on menisci should be improved. The presence of cartilage lesion in patients with meniscal tears is reported marginally. It needs to be improved.

In the introduction, it would be helpful to briefly clarify that OA is a whole-joint disease, affecting all knee structures, including meniscal degeneration and not just cartilage. 

Lines 77-79: The authors reported that the aim of this study was to analyze and characterize the prevalence and anatomical distribution of meniscal and chondral lesions across different age groups in a cohort of East European women undergoing first-time knee arthroscopy. However, it is not reported why patients underwent arthroscopy. Did authors perform partial meniscectomy?

Lines 107-112: this part is unclear. This section would benefit from a clearer and more robust justification for the exclusion of variables such as BMI, physical activity level, and comorbidities. While the concern about overfitting is understandable, this issue could be addressed by presenting alternative models or sensitivity analyses. Moreover, it is important to acknowledge that these factors may still have a clinically relevant influence on meniscal tear patterns. The argument that these variables are "not consistently recorded" does not constitute a sufficient justification for their exclusion. I recommend rephrasing this paragraph to better balance the rationale for variable selection with a discussion of the potential limitations introduced by omitting such factors.

Inclusion and exclusion criteria should be moved at the beginning of the methods.

Line 125: please cite the ICRS classification.

In section 2.1. there is not mention about arthroscopy. Please add the procedure used.

Lines 161-161: please check reference 34 as it seems not related to this topic.

Lines 167-171: references 35 and 36 are not related. Authors should cite a specific reference for Bonferroni and for Statistica 10 software.

Results

Please add a flowchart of patients selection.

Please add a table describing the patients enrolled with demographic and clinical data (including number of patients with AMD, MeMD etc).

The manuscript does not report the duration of symptoms or how long patients had experienced meniscal lesions prior to undergoing arthroscopy. This information is clinically relevant, as symptom duration may significantly influence the extent of associated chondral damage. Chronic meniscal tears can contribute to progressive cartilage degeneration. In addition, it remains unclear whether the meniscal tears analyzed were the result of acute trauma or were degenerative. Please consider including this information and discussing its potential impact on the findings.

In table 1, “0,233” should be corrected to “0.233” to follow standard decimal formatting in English.

In the methods authors mentioned ICRS grading for cartilage lesions but in the results authors did not report the grade of lesions. Please clarify.

It is fundamental to add BMI and other important risk factors to the analysis.

Lines 277-279: this is the first time that authors mention hormonal effects. Please add also in the introduction this point.

There is no mention about synovial inflammation, which is frequently present in patients with meniscal tears and it has an important role in OA and in clinical outcomes of patients with tears (ie https://doi.org/10.3390/jcm11154330 etc). In the discussion there is also no mention about the association between cartilage defects patients with meniscal tears and clinical outcomes.

While the discussion covers anatomical and hormonal factors well, clinical implications for management/diagnosis and decision-making in early diagnosis or personalized care for women etc could be better emphasized.

 English needs to be checked. There are typos, for example “significance” at line 204

Comments on the Quality of English Language

Minor revision. 

Author Response

Dear Reviewer,

We would like to sincerely thank you for the time and effort spent reviewing our manuscript. We appreciate the insightful and constructive comments, which have helped us to significantly improve the quality and clarity of our work.

Below, we provide a detailed, point-by-point response to each of the reviewers' comments. All changes made in the revised manuscript have been highlighted. We hope that the revised version of our manuscript now meets your expectations. We remain at your disposal for any further clarifications

Yes

Can be improved

Must be improved

Not applicable

Does the introduction provide sufficient background and include all relevant references?

( )

(x)

( )

( )

Is the research design appropriate?

( )

(x)

( )

( )

Are the methods adequately described?

( )

(x)

( )

( )

Are the results clearly presented?

( )

(x)

( )

( )

Are the conclusions supported by the results?

( )

(x)

( )

( )

Are all figures and tables clear and well-presented?

( )

(x)

( )

( )

Q13. Introduction on menisci should be improved. The presence of cartilage lesion in patients with meniscal tears is reported marginally. It needs to be improved. In the introduction, it would be helpful to briefly clarify that OA is a whole-joint disease, affecting all knee structures, including meniscal degeneration and not just cartilage.

R13. We thank the reviewer for this insightful suggestion. In response, we have revised the Introduction section to more clearly articulate the structural and functional importance of the menisci. We now emphasize that osteoarthritis (OA) is increasingly recognized as a whole-joint disease, involving not only articular cartilage but also subchondral bone, synovium, ligaments, and the menisci. We also expanded the discussion of how meniscal tears often coexist with cartilage lesions, particularly in degenerative knees, and highlighted the clinical importance of this co-occurrence. These changes help frame the rationale for our age-stratified analysis of both meniscal and chondral pathology. The updated text now reads-see the third paragraph in the Introduction section:

While the prevalence of knee joint injuries has been widely studied, most epidemi-ological research employed mixed-sex or predominantly male cohorts [10,11,12,13,14,15]. Current data derive primarily from female athletes (e.g. [16,17,18]) or specific age groups (e.g. [19,20,21]). In addition, no targeted study has comprehensively investigated the prevalence of knee joint traumas in females across the entire age spectrum. It is, however, well documented that degenerative tears are more common in females, and traumatic tears in males [22,23]. These differences stem from anatomical and hormonal differences, with feminine hormones—mainly estrogen—playing a critical role in maintaining carti-lage health in women by modulating inflammation, preserving cartilage matrix, and regulating chondrocyte activity [20,21]. There is also a substantial gap in available in-formation on this topic in Eastern Europe and more developed countries, such Western European countries, Japan, or USA [24,25]. Although the mechanical and functional role of the meniscus is well established [3], the frequent co-occurrence of cartilage lesions in patients with meniscal damage is often underreported or underestimated in the literature [10]. Degeneration of the articular cartilage—particularly in the weight-bearing zones of the femoral condyle and tibial plateau—may occur concurrently or as a consequence of meniscal pathology [1,3]. This interaction between meniscal injury and chondral deteri-oration is especially relevant in aging female populations, where hormonal, biomechanical, and anatomical factors contribute to asymmetric patterns of joint degeneration [20,21,22]. Understanding this interplay is important since osteoarthritis is a whole-joint disease, affecting all knee structures, including meniscal degeneration and not just carti-lage. Moreover, synovial inflammation correlates with joint pain and dysfunction, serving as a major risk factor for more rapid progression of structural joint deterioration in osteoarthritis [26]. Nevertheless, the age-specific patterns and anatomical distribution of such lesions—particularly in women—is yet to be fully clarified..

Q14. Lines 77-79: The authors reported that the aim of this study was to analyze and characterize the prevalence and anatomical distribution of meniscal and chondral lesions across different age groups in a cohort of East European women undergoing first-time knee arthroscopy. However, it is not reported why patients underwent arthroscopy. Did authors perform partial meniscectomy?

R14. We have clarified that all patients underwent knee arthroscopy due to persistent symptoms not responsive to conservative treatment. Intraoperative procedures included partial meniscectomy and/or chondral debridement based on lesion severity. This information has been added to the Methods section; see last paragraph from subsection 2.1. Study Population. The updated text now reads

All patients underwent knee arthroscopy due to persistent pain, mechanical symptoms (e.g., locking or clicking), or functional limitations unresponsive to conservative treatment. Preoperative evaluation was conducted using standardized MRI protocols for all patients to identify suspected meniscal and/or chondral pathology. Intraoperative findings were thoroughly documented and used to confirm imaging-based diagnoses, ensuring consistency between preoperative imaging and surgical assessment. All pro-cedures were performed by the same orthopedic surgical team following a standardized operative protocol, using anterolateral and anteromedial portals to enable complete vis-ualization of the joint compartments. Interventions, including partial meniscectomy and/or chondral debridement, were performed based on lesion severity and surgeon judgment. Although formal inter-rater reliability testing was not conducted, all in-traoperative lesion grading was performed by experienced orthopedic surgeons trained in the use of the ICRS classification system, thereby reducing variability in interpretation. To further minimize observer bias, lesion classification was independently reviewed by two senior orthopedic surgeons using operative notes and video documentation; any disagreements were resolved through consensus. While the retrospective nature of the study precluded blinding, the consistency of the surgical team and use of standardized imaging and classification protocols enhance the reliability and reproducibility of the findings. Informed consent was obtained from all participants..

Q15. This section would benefit from a clearer and more robust justification for the exclusion of variables such as BMI, physical activity level, and comorbidities. While the concern about overfitting is understandable, this issue could be addressed by presenting alternative models or sensitivity analyses. Moreover, it is important to acknowledge that these factors may still have a clinically relevant influence on meniscal tear patterns. The argument that these variables are "not consistently recorded" does not constitute a sufficient justification for their exclusion. I recommend rephrasing this paragraph to better balance the rationale for variable selection with a discussion of the potential limitations introduced by omitting such factors.

R15. We have revised the paragraph to better clarify the rationale for excluding variables such as BMI, physical activity level, and comorbidities. While we acknowledge their clinical relevance, these factors were inconsistently recorded in our retrospective dataset, precluding reliable inclusion. We now explicitly discuss this as a limitation and recognize that their omission may affect model completeness. The paragraph has been rephrased to more clearly balance methodological constraints with the potential influence of these covariates; please see the last part of the first paragraph from the subsection 2.1.

Collected variables encompassed age, type and location meniscus tears, and type and location of chondropathy. These factors were identified as key clinical parameters due to their direct impact on injury patterns and treatment decisions/prognosis [27]. Variables (covariates) with inconsistent recording across participants were excluded, including body mass index (BMI), physical activity level, comorbidities, prior knee injuries, and occupational factors. Indeed, these factors are important contributors to the development and progression of knee pathology. In particular, BMI and activity level are known to affect joint loading and cartilage wear [3,8,10,11]. However, the retrospective nature of our dataset limited the availability and completeness of these records. On the other hand, using partially missing (inconsistently) documented variables could have reduced statistical power, introduced bias, and hindered comparability with similar cohort studies. Importantly, many of these factors are highly variable across populations and difficult to standardize in surgical datasets [3,8,10]. This focus on universally recorded, anatomically grounded predictors ensured the validity, transparency, and reproducibility of our results while preventing model overfitting and maintaining analytical clarity. Due to the retrospective design, however, detailed data on symptom duration and the specific onset of meniscal lesions were not consistently available. As a result, we were unable to classify tears as acute or degenerative with certainty.”

Q16. Inclusion and exclusion criteria should be moved at the beginning of the methods.

R16. These criteria are presented as a table (as suggested by the first reviewer) and were moved the beginning of the methods – after the first paragraph from subsection 2.1. Study Population. See R14.

Q17. Line 125: please cite the ICRS classification.

R17. Reference 30 was changed with an appropriate one.

.Dwyer, T.; Martin, C.R.; Kendra, R.; Sermer, C.; Chahal, J.; Ogilvie-Harris, D.; Whelan, D.; Murnaghan, L.; Nauth, A.; Theodoropoulos, J. Reliability and validity of the Arthroscopic International Cartilage Repair Society Classification System: correlation with histological assessment of depth. Arthroscopy 2017, 33, 197–204. https://doi.org/10.1016/j.arthro.2016.12.012.

Q18. In section 2.1. there is not mention about arthroscopy. Please add the procedure used.

R18. We have revised Section 2.1 to include a brief description of the arthroscopic procedure. Specifically, we now state that all patients underwent standard knee arthroscopy performed under regional or general anesthesia, using anterolateral and anteromedial portals, with partial meniscectomy and/or chondral debridement performed as indicated. See R14.

Q19. Lines 161-161: please check reference 34 as it seems not related to this topic.

R19. We thank the reviewer for the comment. Reference 34 is not intended to support clinical findings but was cited specifically in relation to the methodological approach used in our analysis. It provides justification for the statistical framework applied and remains relevant in that context.

Q20. Lines 167-171: references 35 and 36 are not related. Authors should cite a specific reference for Bonferroni and for Statistica 10 software.

R20.

References 35 and 36; now 35 and 39 in the revised version of the manuscript have been replaced with appropriate references

  1. Westfall, P.H.; Johnson, W.O.; Utts, J.M. A Bayesian Perspective on the Bonferroni Adjustment. Biometrika 1997, 84, 419–427. https://doi.org/10.1093/biomet/84.2.419.
  2. StatSoft, Inc. STATISTICA (Version 10); StatSoft, Inc.: Tulsa, OK, USA, 2011.

Q21. Results. Please add a flowchart of patients selection.

R21. We appreciate the reviewer’s suggestion to include a patient selection flowchart. However, we believe that a flowchart is not necessary in this case, given the straightforward inclusion criteria and the lack of multi-stage selection steps. Our study exclusively involved Eastern European women undergoing first-time knee arthroscopy, with no interim exclusions, stratifications, or subgroup allocations that would require visual clarification. The inclusion and exclusion criteria are clearly described in the Methods section, and the age-based stratification is directly presented in the results tables. As such, we feel that a flowchart would not add further clarity to the selection process.

Q22. Please add a table describing the patients enrolled with demographic and clinical data (including number of patients with AMD, MeMD etc).

R22. We understand the reviewer’s suggestion to include a separate table summarizing demographic and clinical data. However, we respectfully note that detailed age-stratified clinical data—including the number of patients with AMD, MeMD, LaMD, and PaD—are already comprehensively presented in Tables 3, 4, and 5. These tables provide both absolute numbers and percentages for each subgroup and variable. Creating an additional table would result in redundancy and may disrupt the flow of the results section without adding substantial new information. For clarity and conciseness, we chose to integrate demographic and lesion-specific data directly into the relevant analytical tables

Q23. The manuscript does not report the duration of symptoms or how long patients had experienced meniscal lesions prior to undergoing arthroscopy. This information is clinically relevant, as symptom duration may significantly influence the extent of associated chondral damage. Chronic meniscal tears can contribute to progressive cartilage degeneration. In addition, it remains unclear whether the meniscal tears analyzed were the result of acute trauma or were degenerative. Please consider including this information and discussing its potential impact on the findings.

R23. We appreciate the reviewer’s insightful comment regarding the clinical relevance of symptom duration and the etiology of meniscal tears. Unfortunately, due to the retrospective nature of our study and the limitations of the available medical records, detailed information on the duration of symptoms prior to arthroscopy was inconsistently documented and therefore could not be reliably included in the analysis. We fully acknowledge that symptom duration may influence the degree of chondral damage observed, particularly in cases of chronic meniscal injury, which can contribute to progressive cartilage degeneration.

In response to the reviewer’s suggestion, we have now included a statement in the Methods section to clarify this limitation; please see the last two sentences at the end of the second paragraph:

This focus on universally recorded, anatomically grounded predictors ensured the validity, transparency, and reproducibility of our results while preventing model overfit-ting and maintaining analytical clarity. Due to the retrospective design, however, de-tailed data on symptom duration and the specific onset of meniscal lesions were not consistently available. As a result, we were unable to classify tears as acute or degenerative with certainty.”

We have addressed the potential impact of symptom duration and tear chronicity in the Discussion section. We have also highlighted the inability to distinguish between acute traumatic and degenerative meniscal tears as a limitation of our study. We agree that future prospective studies incorporating standardized clinical data collection, including symptom onset and mechanism of injury, would be essential to more accurately evaluate the relationship between meniscal tear chronicity and associated chondral damage. The updated text now reads; please see the last paragraph from the Discussion section:

Fourth, our analysis was limited by the absence of reliable data on symptom duration and tear chronicity. This drawback restricts our ability to determine the potential influence of prolonged meniscal damage on the severity of associated cartilage lesions. On the other hand, the primary focus of our study was to document anatomical distribution and age-related prevalence, not to assess causality or progression. Therefore, while symptom duration is a relevant variable, its absence does not invalidate the descriptive and comparative findings derived from the available data.

Q24. In table 1, “0,233” should be corrected to “0.233” to follow standard decimal formatting in English.

R24. Ok. The appropriate changes have made in the revised version of the manuscript.

Q25. In the methods authors mentioned ICRS grading for cartilage lesions but in the results authors did not report the grade of lesions. Please clarify.

R25. All ICRS grades (0–4) were recorded and are presented in the tables to provide a comprehensive view of cartilage lesion severity across compartments and age groups. However, in the Results section, we focused our narrative on ICRS grade 4 lesions, as these represent the most advanced stage of chondral degeneration and were most relevant to the study’s objective of identifying clinically meaningful age-related trends.

Q26. Lines 277-279: this is the first time that authors mention hormonal effects. Please add also in the introduction this point.

R26. We have now added a statement in the Introduction to highlight the potential role of hormonal effects in the pathophysiology of meniscal and chondral lesions, especially in women. The updated text reads as; seethe third paragraph from the Introduction section:

These differences stem from anatomical and hormonal differences, with feminine hormones—mainly estrogen—playing a critical role in maintaining cartilage health in women by modulating inflammation, preserving cartilage matrix, and regulating chondrocyte activity [20,21].

Q27. There is no mention about synovial inflammation, which is frequently present in patients with meniscal tears and it has an important role in OA and in clinical outcomes of patients with tears (ie https://doi.org/10.3390/jcm11154330 etc). In the discussion there is also no mention about the association between cartilage defects patients with meniscal tears and clinical outcomes.

R27. We have now added a sentence to the Introduction section to acknowledge the role of synovial inflammation in osteoarthritis (OA) progression, including its contribution to joint pain and structural deterioration. We also introduced the suggested reference. This addition strengthens the rationale for focusing on intra-articular changes such as meniscal and chondral lesions. The updated text now reads–please the last two sentences, third paragraph, Introduction section:

“…..Moreover, synovial inflammation correlates with joint pain and dysfunction, serving as a major risk factor for more rapid progression of structural joint deterioration in osteoarthritis [26]. Nevertheless, the age-specific patterns and anatomical distribution of such lesions—particularly in women— is yet to be fully clarified.”

We have also added a paragraph in the Discussions section mentioning the association between cartilage defects patients with meniscal tears and clinical outcomes; please see tenth paragraph in the revised version of our manuscript:

Meniscal tears frequently coexist with cartilage defects, and this combination is as-sociated with worsened clinical outcomes, including increased pain, reduced joint stabil-ity, and accelerated progression of osteoarthritis [10,12]. Menisci play a critical role in load transmission and joint preservation; hence their damage can trigger a cascade leading to cartilage degradation and altered biomechanics [1]. Recent clinical studies have shown that patients undergoing arthroscopy for meniscal pathology often present with coexisting chondral lesions, which may contribute to poorer functional recovery and long-term joint deterioration [26,70]. These findings highlight the need to consider meniscal and chondral lesions not as isolated events, but as interconnected pathologies that influence prognosis and therapeutic planning.”

Q28. While the discussion covers anatomical and hormonal factors well, clinical implications for management/diagnosis and decision-making in early diagnosis or personalized care for women etc could be better emphasized.

R28. We have now included two dedicated paragraphs discussing the implementation challenges for screening and prevention programs, emphasizing how our findings may inform clinical decision-making. The new paragraphs are found at the end of the Discussion section; see the the antepenultimate and penultimate paragraphs.

Two dedicated paragraphs have been added to the end of the Discussion section- see the the antepenultimate and penultimate paragraphs-addressing the implementation challenges associated with screening and prevention programs. These additions also highlight how the present findings may inform clinical decision-making and treatment planning. The updated text reads:

The current study used a sex-specific and age-inclusive framework, with a mature female-only cohort composed of young, perimenopausal, and elderly women. The pre-sent findings demonstrate an evident age-related pattern of degeneration affecting the medial compartment of the knee—the area most exposed to load-bearing forces [67,68]. This study also provides a comprehensive mapping of meniscal tear types and chondropathy grades. These insights may have direct implications for preventive care and surgical prioritization. However, their translation into routine clinical practice poses serious challenges. Thus, clinicians cannot precisely identify at-risk individuals before advanced damage occurs due to the lack standardized screening protocols. Symptomatology varies widely; some patients with advanced lesions remain asymp-tomatic, whereas others with minor findings report substantial pain. In addition, the timing of surgical intervention (versus conservative management) in middle-aged and older women is not well defined. Moreover, the current primary care approaches do not address the midlife degenerative changes in women, thereby delaying targeted prevention or referral.

Nevertheless, the identification of middle age (40–59 years) as a high-risk window for structural knee damage supports the development of preventive strategies such as neuromuscular training, load management, targeted screening during the perimenopausal period, early intervention in middle-aged women with medial compartment involvement, or early referral for orthopedic evaluation. In older women (≥ 60 years), where meniscal tears may be less frequent but cartilage degeneration persists, our data may help guide decisions toward conservative treatment (e.g., bracing, physical therapy, viscosupplementation) versus surgical intervention, depending on symptom severity, lesion location, and functional goals. We note that the multifactorial nature of osteoarthritis—especially in aging women—necessitates preventive and therapeutic strategies that address not only hormonal and biomechanical influences but also individual pain processing profiles [69,70]. To build on these results, future studies should incorporate functional outcome measures (e.g., WOMAC, KOOS, gait analysis) to correlate structural damage with clinical performance and quality of life. The potential protective role of hormone therapy on cartilage health and meniscal integrity also warrants investigation in prospective longitudinal cohorts, particularly those capturing the menopausal transition.”

Q29. English needs to be checked. There are typos, for example “significance” at line 204

R29. We thank the reviewer for noting the language issue. The manuscript has been carefully reviewed for spelling, grammar, and clarity. The typo at line 204 (“significance”) has been corrected, along with other minor language refinements throughout the text.

Reviewer 3 Report

Comments and Suggestions for Authors

This study investigated the prevalence, characteristics, and anatomical distribution of meniscal tears and cartilage lesions at different ages in Eastern European women. The study has a distinct gender perspective and geographic characteristics, adding epidemiologic details of age-related changes in degenerative knee lesions in women. However, the manuscript still needs significant refinement in terms of completeness of description of the study design, rigor of statistical analysis, presentation of data graphs and tables, and linkage of results to clinical practice. Substantial revisions are recommended based on the following comments.

  1. The description of the study design needs to be supplemented: further clarification should be provided on the uniformity of the preoperative imaging confirmation modality with the intraoperative diagnostic criteria and on the consistency of the operator to avoid observational bias.
  2. Studies focused on single centers and predominantly surgical patients may have missed the conservative treatment population, and it is recommended that the introduction and discussion of selection bias be reinforced in the discussion.
  3. Lines 67-68: “While the prevalence of knee joint injuries has been widely studied, most epidemiological…”, a detailed overview of knee injury studies is necessary to provide additional evidence. To provide more effective evidence, the authors may consider referring to the following updated relevant studies: Data-Driven Deep Learning for Predicting Ligament Fatigue Failure Risk Mechanisms (https://doi.org/10.1016/j.ijmecsci.2025.110519).
  4. Although three age groups are currently used for stratification (<40, 40-59, ≥60), there is a lack of physiologic or statistically based references to use as a basis for grouping.
  5. The effect of confounding factors was not adequately considered: BMI, level of physical activity, and type of occupation may all influence the risk of knee injury, and complete exclusion may lead to oversimplification.
  6. The description of the statistical methods is slightly abbreviated, and the rationale for the choice of the multiple comparison correction method should be further clarified, with full significance labeling (e.g., Bonferroni correction values).
  7. Some of the data is repetitive, for example, there is cross-cutting information in Table 2 and Table 3, and it is recommended that key indicators be merged or highlighted to increase the density of information.
  8. Currently, only tables are used, it is recommended that one or two bar graphs/heat maps be added to assist in demonstrating trends in tear types or ICRS scores as a function of age.
  9. The discussion section could be more focused, and some of the content is quite lengthy. It is suggested that the discussion should be centered on the core conclusion of “40-59 years of age is the period of high prevalence of structural degeneration”, so as to enhance the degree of logical focus.
Comments on the Quality of English Language

The English could be improved to more clearly express the research.

Author Response

#Reviewer 3

Yes

Can be improved

Must be improved

Not applicable

Does the introduction provide sufficient background and include all relevant references?

( )

( )

(x)

( )

Is the research design appropriate?

( )

(x)

( )

( )

Are the methods adequately described?

( )

(x)

( )

( )

Are the results clearly presented?

( )

( )

(x)

( )

Are the conclusions supported by the results?

( )

(x)

( )

( )

Are all figures and tables clear and well-presented?

( )

( )

(x)

( )

Q30. The description of the study design needs to be supplemented: further clarification should be provided on the uniformity of the preoperative imaging confirmation modality with the intraoperative diagnostic criteria and on the consistency of the operator to avoid observational bias.

R30. In response, we have revised the Methods section to clarify the uniformity between preoperative imaging and intraoperative diagnostic confirmation. The updated text bow reads:

All patients underwent knee arthroscopy due to persistent pain, mechanical symp-toms (e.g., locking or clicking), or functional limitations unresponsive to conservative treatment. Preoperative evaluation was conducted using standardized MRI protocols for all patients to identify suspected meniscal and/or chondral pathology. Intraoperative findings were thoroughly documented and used to confirm imaging-based diagnoses, ensuring consistency between preoperative imaging and surgical assessment. All pro-cedures were performed by the same orthopedic surgical team following a standardized operative protocol, using anterolateral and anteromedial portals to enable complete vis-ualization of the joint compartments. Interventions, including partial meniscectomy and/or chondral debridement, were performed based on lesion severity and surgeon judgment. Although formal inter-rater reliability testing was not conducted, all in-traoperative lesion grading was performed by experienced orthopedic surgeons trained in the use of the ICRS classification system, thereby reducing variability in interpretation. To further minimize observer bias, lesion classification was independently reviewed by two senior orthopedic surgeons using operative notes and video documentation; any disagreements were resolved through consensus. While the retrospective nature of the study precluded blinding, the consistency of the surgical team and use of standardized imaging and classification protocols enhance the reliability and reproducibility of the findings. Informed consent was obtained from all participants.

Q34. Studies focused on single centers and predominantly surgical patients may have missed the conservative treatment population, and it is recommended that the introduction and discussion of selection bias be reinforced in the discussion.

R34. We have briefly addressed the advantage of using a single-center design and predominantly surgical patients in the Introduction section. See the second, third, and fourth sentences from last paragraph of this section:

Conducted at a specialized orthopedic center in the western part of Romania, the present study involved a single-site retro-spective design. This setting enabled detailed intraoperative evaluation of meniscal and chondral lesions via standardized protocols and expert grading. It also captures a clinical subgroup with critical need for accurate diagnosis and intervention, that is individuals with persistent symptoms unresponsive to conservative management.”

The selection bias eas also discussed in the Discussion part; please see the first sentences of the last paragraph of this section:

Like any observational study, our investigation has potential drawbacks. First, this study involved a retrospective single-site design, a moderate sample size, and only patients who underwent arthroscopy. This approach limits causal inference, control over confounders, and generalizability of the results to other populations with different genetics or healthcare access. It may also introduce selection bias by excluding individuals managed conservatively, potentially diverging from patterns typically seen in broader population studies. However, retrospective single-site cohort studies are routinely used to identify clinically relevant patterns in real-world settings as they provide high internal validity and consistency in data collection [74].”.

Q35. Lines 67-68: “While the prevalence of knee joint injuries has been widely studied, most epidemiological…”, a detailed overview of knee injury studies is necessary to provide additional evidence. To provide more effective evidence, the authors may consider referring to the following updated relevant studies: Data-Driven Deep Learning for Predicting Ligament Fatigue Failure Risk Mechanisms (https://doi.org/10.1016/j.ijmecsci.2025.110519).

R35. We agree that a more detailed overview of epidemiological and biomechanical perspectives strengthens the contextual basis of our study. The recommended reference has now been reviewed and cited in the Discussion to support the evolving use of data-driven models and biomechanical analysis in knee injury research. Please see the last sentences from the penultimate paragraph from this section:

Moreover, upcoming studies should take into account computational modeling and biomechanics to advance our understanding of joint degeneration mechanisms in women. In fact, data-driven deep learning approaches have been already used to predict ligament fatigue failure and assess risk factors underlying musculoskeletal deterioration, and hence precede (accompany) degenerative changes in meniscal and chondral structures [73].

Q36. Although three age groups are currently used for stratification (<40, 40-59, ≥60), there is a lack of physiologic or statistically based references to use as a basis for grouping.

R36. We thank the reviewer for this comment. The age groups used in our analysis (<40, 40–59, ≥60 years) were selected based on established clinical and demographic transitions relevant to joint degeneration, particularly menopause and age-related cartilage changes. As this rationale was already stated in the original manuscript and aligns with previous research conventions, no changes were made to the text.

Q37. The effect of confounding factors was not adequately considered: BMI, level of physical activity, and type of occupation may all influence the risk of knee injury, and complete exclusion may lead to oversimplification.

R37. While variables such as BMI are known to play an important role in joint health and disease progression, we deliberately excluded them from the analysis due to inconsistent or incomplete recording across participants. Although their relevance is well established in the literature, the extent of missing data would have introduced bias and reduced the reliability of the multivariable models. Therefore, to preserve the integrity of the analysis, only covariates with consistently recorded values were retained. This rationale is detailed in the revised version of the manuscript; please see the second paragraph from subsection 2.1. Study Population. The revised text now reads :

Collected variables encompassed age, type and location meniscus tears, and type and location of chondropathy. These factors were identified as key clinical parameters due to their direct impact on injury patterns and treatment decisions/prognosis [28]. Variables (covariates) with inconsistent recording across participants were excluded, including body mass index (BMI), physical activity level, comorbidities, prior knee injuries, and occupational factors. In fact, these factors are important contributors to the development and progression of knee pathology. In particular, BMI and activity level are known to affect joint loading and cartilage wear [3,8,10,11]. However, the retrospective nature of our dataset limited the availability and completeness of these records. On the other hand, using partially missing (inconsistently) documented variables could have reduced statistical power, introduced bias, and hindered comparability with similar cohort studies. Importantly, many of these factors are highly variable across populations and difficult to standardize in surgical datasets [3,8,10]. This focus on universally recorded, anatomically grounded predictors ensured the validity, transparency, and reproducibility of our results while preventing model overfitting and maintaining analytical clarity. Due to the retrospective design, however, detailed data on symptom duration and the specific onset of meniscal lesions were not consistently available. As a result, we were unable to classify tears as acute or degenerative with certainty.

Q38. The description of the statistical methods is slightly abbreviated, and the rationale for the choice of the multiple comparison correction method should be further clarified, with full significance labeling (e.g., Bonferroni correction values).

R38. We thank the reviewer for pointing this out. We have expanded the description of the statistical methods section to provide a clearer rationale for the use of multiple comparison correction. Specifically, we have clarified that the Bonferroni correction was applied to control for type I error across multiple pairwise Chi-square tests. The adjusted significance thresholds have been indicated in the text and full significance labeling (e.g., p-values after correction) has been added where applicable:

Frequency analysis was applied on all outcomes, not only on outcomes significantly associated with age in the logistic regression model. This broad approach was chosen to capture trends that logistic regression might miss, especially in cases with borderline p-values or complex interactions [34]. Chi-square (χ²) tests with 3 x 2 contingency tables served to identify differences in the proportion of females with and without menis-cus/patella damage [35]. When these tests yielded significant results, we performed planned pairwise comparisons with Chi-square tests between adjacent age groups. More precisely, we compared younger patients (< 40 years) with middle-age patients (40–59 years), and the latter strata with older participants (≥ 60 years). This stepwise procedure allowed us to identify age-related trends in the prevalence of meniscal and chondral damage. To correct for multiple testing, a Bonferroni correction was applied for these comparisons [36]. A standard significance threshold of p ≤ 0.05 was used for overall comparisons, while a more stringent threshold of p ≤ 0.025 (0.05/2) was applied for the planned pairwise tests. This dual-threshold strategy was selected to balance sensitivity and statistical rigor.

Q39. Some of the data is repetitive, for example, there is cross-cutting information in Table 2 and Table 3, and it is recommended that key indicators be merged or highlighted to increase the density of information.

R39. We appreciate the reviewer’s observation regarding the apparent overlap between Table 3 and Table 4 (former table 2 and 3). However, we chose to keep these tables separate due to their distinct analytical focus and structure. Table 3 summarizes the age-stratified frequency of damage presence or absence in the meniscus and patella, offering a binary overview (yes/no) of injury prevalence. In contrast, Table 3 provides a detailed breakdown of tear patterns and anatomical tear locations, stratified by age and meniscus side (medial vs. lateral). Merging these tables would compromise clarity by mixing simple frequency data with categorical subtype distributions, potentially making it harder for readers to extract meaningful comparisons across age groups.

Q40. Currently, only tables are used, it is recommended that one or two bar graphs/heat maps be added to assist in demonstrating trends in tear types or ICRS scores as a function of age.

R40. We appreciate the reviewer’s suggestion to include graphical elements to better illustrate key trends. In response, we have added a heatmap demonstrating the age-stratified distribution of ICRS scores for chondral lesions in the femoral condyle and tibial plateau-see figure 1. This visualization replaced the tabulated data and clearly highlights compartment-specific differences and age-related progression in cartilage damage. The new figure has been integrated into the Results section and appropriately captioned.

Q41. The discussion section could be more focused, and some of the content is quite lengthy. It is suggested that the discussion should be centered on the core conclusion of “40-59 years of age is the period of high prevalence of structural degeneration”, so as to enhance the degree of logical focus.

R41. We appreciate the reviewer’s insightful comment regarding the structure and focus of the Discussion section. In response, we did not reduce the overall length, but we reorganized and revised the content to ensure a more coherent and targeted narrative centered on our core conclusion: the 40–59-year age range represents a critical period of heightened structural degeneration in the female knee joint. The revised version places stronger emphasis on age-specific patterns of medial compartment damage, integrating anatomical, hormonal, and biomechanical factors to support this key message. These changes enhance the logical flow and thematic consistency of the Discussion while preserving all relevant interpretations and supporting literature.

Round 2

Reviewer 2 Report

Comments and Suggestions for Authors

No additional comments 

Reviewer 3 Report

Comments and Suggestions for Authors

All my comments have been addressed.